# Survival Rate after Palliative Surgery Alone for Symptomatic Spinal Metastases: A Prospective Cohort Study

**DOI:** 10.3390/jcm11216227

**Published:** 2022-10-22

**Authors:** Kenichiro Kakutani, Yoshitada Sakai, Zhongying Zhang, Takashi Yurube, Yoshiki Takeoka, Yutaro Kanda, Kunihiko Miyazaki, Hiroki Ohnishi, Tomoya Matsuo, Masao Ryu, Kohei Kuroshima, Naotoshi Kumagai, Yoshiaki Hiranaka, Shinya Hayashi, Yuichi Hoshino, Hitomi Hara, Ryosuke Kuroda

**Affiliations:** 1Department of Orthopaedic Surgery, Kobe University Graduate School of Medicine, 7-5-1 Kusunoki-cho, Chuo-ku, Kobe 650-0017, Japan; 2Division of Rehabilitation Medicine, Kobe University Graduate School of Medicine, 7-5-1 Kusunoki-cho, Chuo-ku, Kobe 650-0017, Japan

**Keywords:** symptomatic spinal metastases, spine surgery, prognosis

## Abstract

The effect of spine surgery for symptomatic spinal metastases (SSM) on patient prognosis remains unclear. This study aimed to reveal the prognosis of patients with SSM after spine surgery. One hundred twenty-two patients with SSM were enrolled in this prospective cohort study. The patients who received chemotherapy after enrollment were excluded. The decision of surgery depended on patient’s willingness; the final cohort comprised 31 and 24 patients in the surgery and non-surgery groups, respectively. The patients were evaluated by their performance status (PS), activities of daily living (ADL) and ambulatory status. Survival was evaluated by the Kaplan–Meier method. The PS, ADL and ambulation were significantly improved in the surgery group compared to non-surgery group. The median survival was significantly longer in the surgery group (5.17 months, 95% confidence interval (CI) 3.27 to 7.07) than in the non-surgery group (2.23 months, 95% CI 2.03 to 2.43; *p* = 0.003). Furthermore, the patients with a better PS, ADL and ambulatory status had a significantly longer survival. Surgery improved the PS, ADL, ambulation and survival of patients with SSM. In the management of SSM, spine surgery is not only palliative but may also prolong survival.

## 1. Introduction

The incidence of bone/spinal metastasis has increased because of improvements in cancer therapy [1,2,3,4]. Bone/spinal metastasis is a debilitating complication that causes intractable pain and/or neurological deficits [5]. In particular, a symptomatic spinal metastasis (SSM) markedly decreases the performance status (PS) and activities of daily living (ADL) [6], rendering it difficult to maintain and improve the quality of life (QOL). Furthermore, because SSM decreases the PS and ADL, primary cancer treatments such as chemotherapy and radiotherapy must often be cancelled due to the patients’ poor general condition. Therefore, patient survival is shortened. Bone/spinal metastasis is thus a growing global health problem that requires appropriate management.

In the management of SSM, Patchell et al. reported that direct decompressive surgery followed by radiotherapy is superior to radiotherapy alone in improving patients’ ambulatory status and survival rate [7]. However, in contrast to these findings, a matched pair analysis did not prove the superiority of spine surgery over radiotherapy [8]. Other studies revealed that spine surgery improves physical activity, pain, and neurological function [9,10]. Prospective cohort studies have also investigated the effect of spine surgery on the PS, ADL, neurological status and QOL of patients with SSM [6,11,12]; however, these studies included patients who underwent postoperative chemotherapy, which affects survival. Overall, the impact of spine surgery for SMM remains unclear and requires investigation in a prospective study of patients who do not undergo postoperative chemotherapy.

At present, the management of SSM is ultimately palliative and aims to achieve a favorable QOL. In addition, it is important for patients with SSM to stay at home as long as possible until they reach the terminal phase. The prediction of the chronological clinical course of patients with SSM is essential for the planning of a multidisciplinary treatment or terminal care.

Studies have reported patient survival rate after surgery for SSM and retrospectively identified independent prognostic factors. The ambulatory status is strongly associated with prolonged survival [13,14,15]. In addition, the body mass index is an independent predictor of survival, as a favorable nutritional status is associated with improved survival [16]. Thus, surgery is expected to indirectly improve survival. However, Jansson and Bauer [10] concluded that surgery for SSM is purely palliative, does not affect survival and must be weighed against other treatment options. The impact of spine surgery on the survival and outcome of patients with SSM remains controversial. We hypothesized that spine surgery improves the general condition, ambulatory status, ADL and QOL of patients with SSM, and that these effects improve survival. Hence, this prospective cohort study aimed to determine the effect of surgery on the survival rate and outcomes of patients with SSM.

## 2. Patients and Methods

### 2.1. Ethics Statement

This prospective cohort study was conducted at our hospital. The study protocol was approved by the ethics committee and institutional review board of our hospital. Written informed consent was obtained from each patient. The study was conducted in concordance with the principles of the Declaration of Helsinki and with the laws and regulations of our country.

### 2.2. Patients and Procedures

This prospective cohort study was performed in a single institution. A consecutive cohort of 122 patients with SSM was prospectively studied from 2013 to 2017. The diagnosis of spinal metastasis was based on magnetic resonance imaging and computed tomography of the whole spine. In some cases, the final diagnosis was established using positron emission tomography–computed tomography and needle biopsy.

SSM was defined as spinal metastases associated with progressive neurological deficits, spinal instability or intractable pain resistant to conservative care. Consequently, all patients with SSM were surgical candidates. The exclusion criteria were (1) impaired consciousness due to cerebral metastasis, (2) terminal phase with an expected survival of maximum 2 weeks, (3) complete paraplegia for more than 72 h, (4) previous spine surgery, (5) a single spinal metastasis lesion (such patients received curative treatment via total spondylectomy).

After the diagnosis of SSM, a recommendation of spine surgery and a comprehensive explanation of the risks and benefits of surgery and other adjuvant therapies were given to all patients and their families using written documentation. Patients chose to undergo surgery by their own volition. Both patients who underwent surgery (surgery group) and those who did not (non-surgery group) were administered radiotherapy, physical therapy and palliative care services. Chemotherapy, including molecular targeted drugs and hormone therapy, was administered if indicated. However, as chemotherapy is a strong positive prognostic factor, patients who underwent chemotherapy were excluded from the present study in order to maintain the homogeneity of the cohort and enable the evaluation of the effect of spine surgery alone on the outcome (Figure 1). The patients underwent clinical evaluations before treatment.

All surgeries involved single-stage posterior decompression and stabilization with fixation using lateral mass screws for the cervical spine and pedicle screws for the thoracic and lumbar spine. Neither corpectomy nor an anterior approach was performed. All patients then underwent the removal of their immobilization devices. The patients were mobilized immediately after surgery and underwent radiotherapy and chemotherapy 2 weeks postoperatively.

### 2.3. Clinical Assessment

The clinical evaluations included the Tokuhashi score [17], the new Katagiri score [18,19], a nomogram [20], the Eastern Cooperative Oncology Group PS [21], the Barthel Index (BI) [22], the ambulatory status based on the Frankel classification [23] and the EuroQol Five Dimension questionnaire score (EQ-5D) [24]. A BI of less than 40 points indicates the need for another person’s help for every daily activity. The primary endpoint was the PS, BI, QOL and ambulatory status at 1 month after the diagnosis of SSM. Follow-up was routinely performed once a month from the study commencement until patient’s death. The survival time after study commencement was recorded as the final endpoint. Patients who were alive but could not visit our department were contacted by telephone to obtain the final follow-up information [25].

### 2.4. Statistical Analysis

All values are expressed as mean ± standard deviation. Demographic data were compared using the t-test, Mann–Whitney U test, chi-square test, and Fisher exact test. The survival rates of the surgery and non-surgery groups were compared by the Kaplan–Meier method and log-rank test.

The potential prognostic factors investigated at study commencement in both groups were: sex, age, Tokuhashi score, new Katagiri score, nomogram, location, primary cancer growth rate, visceral or cerebral metastasis, use of a bone-modifying agent (BMA), PS, ambulatory status, BI and EQ-5D. The location was categorized based on the spinal instability neoplastic score as junctional spine (occiput–C2, C7–T2, T11–L1, L5–S1), mobile spine (C3–6, L2–4), or rigid spine (T3–10) [26]. The primary cancer was categorized using the new Katagiri score as a slowly growing tumor (hormone-dependent breast and prostate cancer, thyroid cancer, multiple myeloma, malignant lymphoma), a moderately growing tumor (lung cancer treated with molecular targeted drugs, hormone-independent breast and prostate cancer, renal cell carcinoma, endometrial and ovarian cancer, sarcoma, others) or a rapidly growing tumor (lung cancer not treated with molecular targeted drugs, colorectal cancer, esophageal cancer, other urological cancers, hepatocellular carcinoma, gall bladder cancer, cervical cancer, cancer of unknown origin) [18]. Categorical variables with more than two modalities were recorded using dummy variables.

For the chronological evaluation of the endpoints, the Kruskal–Wallis test and Scheffe post hoc test were used to assess the significance of differences between the surgery and the non-surgery groups. An intention-to-treat population was used throughout. Statistical significance was set at *p* < 0.05. The analyses were performed using PASW Statistics 18 (SPSS, Chicago, IL, USA).

## 3. Results

### 3.1. Patient Demographics and Baseline Clinical Characteristics

One hundred and twenty-two patients (mean age, 66.0 ± 11.6 years; range 32 to 89 years) were enrolled. Based on the patients’ wishes, 86 patients underwent surgery, while 36 received conservative treatment. Chemotherapy was subsequently administered to 55 patients in the surgery group and 12 in the non-surgery group; these patients were excluded from this study. The final cohort comprised 31 patients in the surgery group and 24 in the non-surgery group (Figure 1).

In the surgery group, the mean number of stabilized levels was 6.6 ± 1.8 (range 3 to 10), the mean surgical time was 198.9 ± 62.9 min (range 77 to 370 min), and the mean intraoperative bleeding volume was 406.4 ± 513.7 mL (range 10 to 2500 mL). Postoperative complications occurred in five patients (16.1%), comprising acute surgical site infection, acute myocardial infarction, severe peritonitis, ureteral stenosis and a hydrocephalus-induced consciousness disorder; each complication occurred in one patient. All patients with postoperative complications underwent additional planned therapies. No patients required additional surgery for spinal metastasis at another spinal level. All patients received radiotherapy with a mean dose of 29.5 ± 2.9 Gy (range 15 to 34 Gy) within 2 weeks after the study commencement.

There were no significant differences between the surgery and the non-surgery groups regarding the Tokuhashi’s score, new Katagiri’s score, nomogram, PS, BI and EQ5D at baseline (Table 1). The most frequent types of primary cancer were sarcoma and hepatocellular carcinoma (Table 2). There were no significant differences between the surgery and the non-surgery groups regarding the malignancy grade in accordance with the new Katagiri’s score, cancer location and presence of visceral or cerebral metastasis at baseline (Table 1). The prevalence of ambulation was 48.4% in the surgery group and 70.8% in the non-surgery group (*p* = 0.08).

### 3.2. Chronological Changes in Clinical Characteristics

The median PS of the non-surgery group was 3 (range 2 to 4) at baseline, 4 (range 2 to 4) at 1 month after study commencement, and 4 (range 4 to 4) at 6 months. The PS of the surgery group was instantly improved to 2 (range 1 to 4) at 1 and 3 months postoperatively. The PS was significantly better in the surgery group than in the non-surgery group at 1 month (*p* < 0.001) and 3 months (*p* < 0.001). However, at the final endpoint, the median PS of the surgery group had deteriorated to 3 (range 1 to 4) and was not significantly different from that of the non-surgery group (*p* = 0.08) (Figure 2).

The mean BI of the non-surgery group was 48.8 ± 31.2 (range 5 to 100) at baseline, 37.9 ± 32.0 (range 0 to 100) at 1 month, 30.4 ± 23.5 (range 5 to 75) at 3 months and 15.0 ± 5.0 (range 10 to 20) at 6 months. In contrast, the BI of the surgery group was instantly increased to 73.5 ± 26.5 (range 10 to 100) at 1 month postoperatively and was maintained at more than 70 points until 3 months after the study commencement; the mean BI was significantly greater in the surgery group than in the non-surgery group at 1 month (*p* < 0.001) and 3 months (*p* < 0.001). The mean BI in the surgery group remained above 70 points until 6 months postoperatively but was similar to that in the non-surgery group at the final endpoint (*p* = 0.148) (Figure 3). Similar trends were seen for QOL and the EQ5D. The mean EQ5D was significantly and instantly improved by surgery until 3 months (*p* < 0.001 versus the non-surgery group at 1 and 3 months) but decreased at 6 months (*p* = 0.057 versus the non-surgery group) (Figure 4).

The prevalence of ambulatory patients was significantly higher in the surgery group than in the non-surgery group at 1 month (*p* = 0.007) and 3 months (*p* = 0.028) but not at 6 months (*p* = 0.345) (Table 3).

### 3.3. Overall Survival Time

The respective survival rates in the surgery and non-surgery groups were 93.5% and 79.2% at 1 month, 67.7% and 29.2% at 3 months, and 22.6% and 8.3% at 6 months. The median survival time (MST) was significantly longer in the surgery group (5.17 months, 95% confidence interval (CI) 3.27 to 7.07 months) than in the non-surgery group (2.23 months, 95% CI 2.03 to 2.43 months; *p* = 0.003) (Figure 5).

The cause of death in the non-surgery group was the primary cancer in all patients. In the surgery group, two patients were suspected to have died of surgery-related complications, comprising a subacute surgical site infection (*n* = 1) and a severe peritonitis (*n* = 1). The survival time was 3.97 months for the patient with subacute surgical site infection and 3.53 month for the patient with severe peritonitis. The other patients in the surgery group died of their primary cancers. Two patients unexpectedly died of acute exacerbation of a lung metastasis within 1 month postoperatively; one was a patient with osteosarcoma who had a new Katagiri’s score of 7, a Tokuhashi’s score of 6, and a nomogram of 370, while the other was a patient with cancer of unknown origin with a new Katagiri’s score of 4, a Tokuhashi’s score of 8, and a nomogram of 330.

Regarding the relationship between the survival rate and the PS, ambulation and ADL at 1 month after the study commencement, the MST of patients with a PS of 0–2 (in both the surgery and the non-surgery groups) was significantly extended compared with that of patients with a PS of 3–4 (*p* = 0.004) (Table 4). In addition, the MST was significantly longer for ambulatory patients than for non-ambulatory patients (*p* = 0.009) (Table 5) and for patients with a BI ≥ 60 compared with patients with a BI < 60 (*p* = 0.029) (Table 6).

## 4. Discussion

This prospective study was performed to elucidate the effect of spine surgery after the development of SSM. The results revealed that spine surgery for SSM significantly improved the PS, ADL, ambulatory status and QOL. In addition, patients who underwent spine surgery had a significantly longer survival than those who did not undergo surgery.

The natural history of patients with spinal cord compression due to spinal metastasis remains unclear [27]. In the current study, 24 patients did not undergo either surgery or chemotherapy; in this group, the MST was 2.23 months, and the PS, BI, EQ-5D and ambulatory status decreased during 6 months of follow-up. These results suggest that the natural history of SSM is severe deterioration. In contrast, spine surgery instantly improved the PS, BI, EQ-5D and ambulatory status, and these improvements were maintained for 3 months. However, the PS and EQ-5D regressed at 6 months. As the improvements in the BI, ambulatory status and neurological status were maintained for 6 months, the main reason for the re-exacerbation of the PS and EQ-5D might be the progression of the primary cancer rather than the exacerbation of spinal cord compression.

It remains unclear whether the current management options improve the condition of patients with SSM. Studies on spinal metastasis have reported that radiotherapy improves pain [28,29], and BMA therapy prevents skeletal-related events [30]. However, there is no evidence that radiation and BMA therapy improve intractable pain and deteriorated neurological status due to SSM. Furthermore, the previous studies that reported that spine surgery improves spinal instability and neurological status were retrospective, and the results included the effect of chemotherapy. Thus, the current study was performed to evaluate the effect of spine surgery without chemotherapy on the PS, ADL, walking ability and QOL of patients with SSM.

Regarding the effect of spine surgery for SSM on survival, several studies report that spine surgery provides the opportunity for patients to receive adjuvant therapy and thus may lead to improved survival [9,13,31]. However, these studies were also retrospective and included patients who received chemotherapy. Therefore, there is a need for a prospective study to investigate the effect of spine surgery alone (i.e., excluding patients who receive chemotherapy) on the survival of patients with SSM. In the current study, the survival time was relatively short in the non-surgery group but was significantly extended in the surgery group. As there were no significant differences between the two groups in baseline clinical characteristics and adjuvant therapy, this result suggests that spine surgery improves the survival of patients with SSM.

To determine the relationship between survival and general condition, we assessed the PS, ambulatory status and BI using the log-rank test. The factors associated with a significantly longer survival were a PS of 0–2, an ambulatory neurological status and a BI ≥ 60. As the spine is not a vital organ, spine surgery does not directly improve the survival. Therefore, the longer survival after spine surgery might be due to improvements in patients’ general condition. Furthermore, many enrolled patients who underwent surgery were subsequently excluded because they received chemotherapy. These patients were able to undergo chemotherapy because spine surgery improved their PS and ADL. As chemotherapy is a strong prognostic factor, the survival of patients who received both spine surgery and chemotherapy is expected to be much longer than that of patients who received spine surgery alone. There is a need for a future study evaluating multidisciplinary treatments for SSM.

The major limitations of the current study are the small sample size and the lack of randomization regarding the treatment. As patients decided to undergo surgery by their own volition, the results may have been affected by their positive feelings associated with the completion of treatment (i.e., patients may have chosen surgery to receive chemotherapy). This might have affected the improvements in the PS, ADL and ambulation, leading to the administration of chemotherapy. The exclusion of patients who received chemotherapy means that the study cohort comprised patients with sarcoma or hepatocellular carcinoma and those with contraindications for chemotherapy. Thus, the results might be limited to patients with specific types of cancer that rarely cause bone metastasis.

In summary, this prospective cohort study revealed that spine surgery alone improves the general condition and survival rate of patients with SSM.

## Figures and Tables

**Figure 1 jcm-11-06227-f001:**
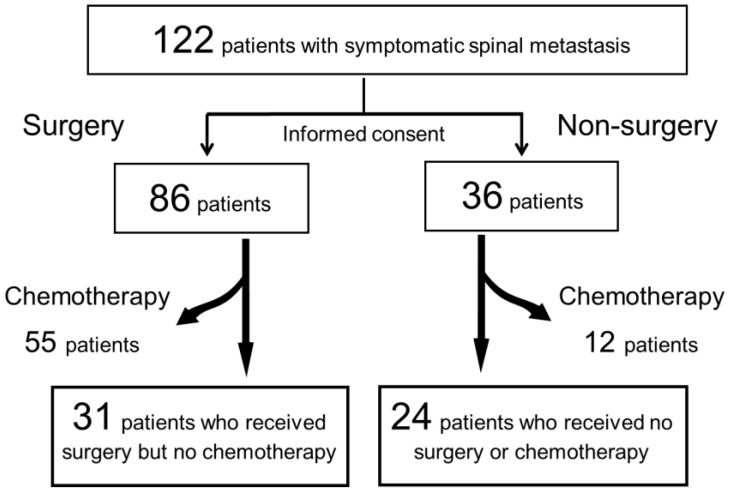
Flowchart of the study design.

**Figure 2 jcm-11-06227-f002:**
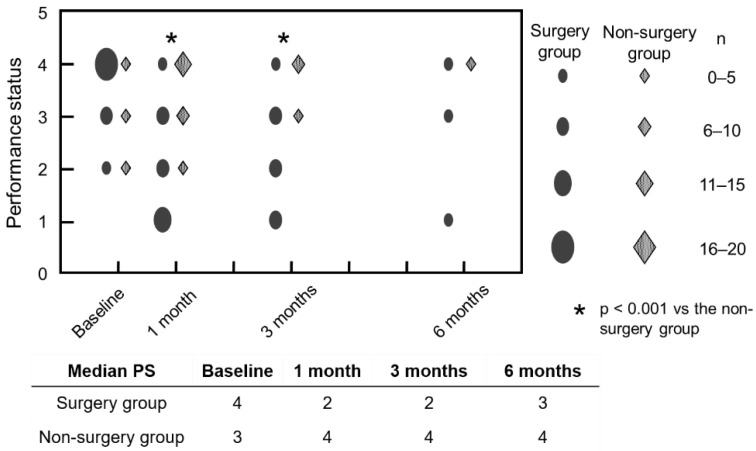
Chronological changes in the PS preoperatively and at 1, 3 and 6 months after study commencement. PS: performance status.

**Figure 3 jcm-11-06227-f003:**
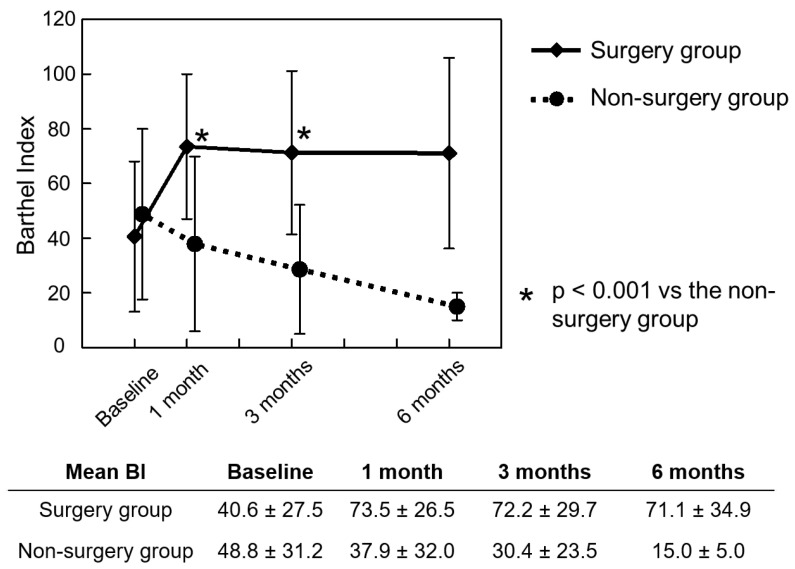
Chronological changes in the BI preoperatively and at 1, 3 and 6 months after study commencement; BI: Barthel Index.

**Figure 4 jcm-11-06227-f004:**
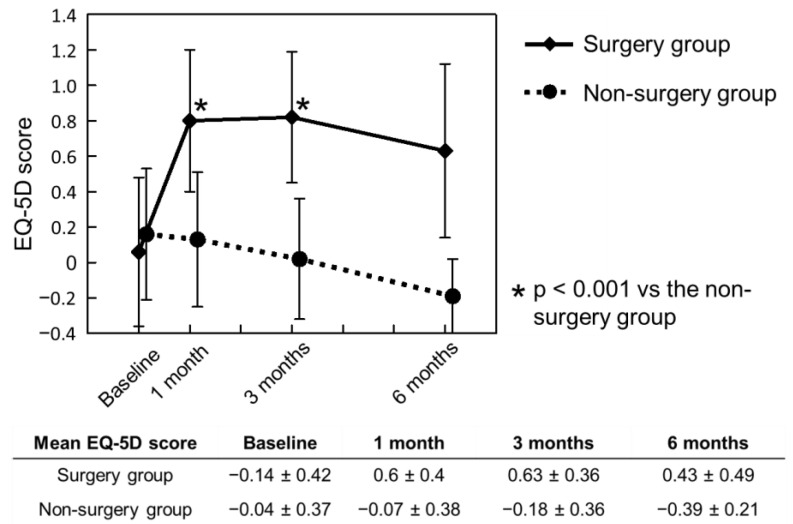
Chronological changes in the EQ-5D score preoperatively and at 1, 3 and 6 months after study commencement; EQ-5D: EuroQol Five-Dimension questionnaire.

**Figure 5 jcm-11-06227-f005:**
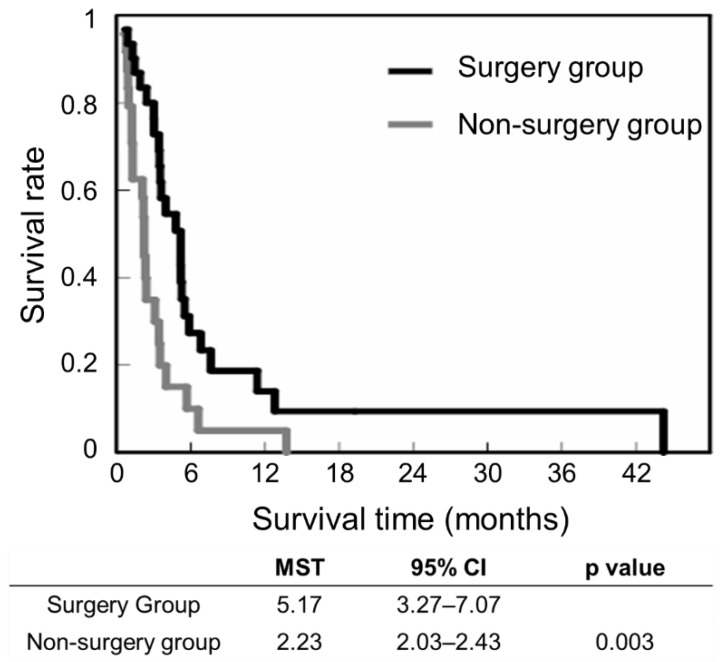
Kaplan–Meier survival rate; CI: confidence interval; MST: median survival time.

**Table 1 jcm-11-06227-t001:** Baseline demographics and clinical characteristics of patients who underwent surgery and of those who did not.

Background Characteristic	Subgroup	Surgery Group	Non-Surgery Group	*p*-Value
Sex, *n*	Male	21	16	
	Female	10	8	0.580
Age, mean ± SD		70 ± 9.9	69 ± 13.6	0.766
Tokuhashi score, mean ± SD		5.3 ± 2.5	5.0 ± 1.4	0.618
New Katagiri score, mean ± SD		5.8 ± 1.7	6.4 ± 1.1	0.150
Nomogram, mean ± SD		332.3 ± 57.4	318.8 ± 44.7	0.346
Location, *n*	Rigid spine (reference group)	12	8	
	Mobile spine	8	7	
	Junctional spine	11	9	0.915
Primary cancer growth rate, *n*	Slow growth (reference group)	3	0	
	Moderate growth	14	16	
	Rapid growth	14	8	0.139
Visceral or cerebral metastasis, *n*	Yes	23	17	
	No	8	4	0.509
Bone-modifying agent therapy, *n*	Yes	13	13	
	No	18	11	0.265
PS 2, *n*		4	6	
PS 3, *n*		8	8	
PS 4, *n*		19	10	
Median PS		4	3	0.130
Ambulation, *n* (%)	Yes	15 (48.8)	17 (70.8)	0.080
Barthel Index, mean ± SD		40.6 ± 27.9	48.7 ± 31.9	0.321
EQ-5D score, mean ± SD		−0.1 ± 0.42	0.0 ± 0.37	0.167

SD: standard deviation; PS: performance status.

**Table 2 jcm-11-06227-t002:** Type of primary cancer.

Primary Lesion	Number of Patients	
	Surgery Group	Non-Surgery Group
Lung cancer	5	5
Sarcoma	5	3
Hepatocellular carcinoma	4	4
Renal cell carcinoma	0	4
Colon and rectal cancer	2	1
Other urological cancer	2	1
Bladder cancer	1	2
Esophageal cancer	2	0
Thyroid cancer	2	0
Others	2	0
Breast cancer	1	0
Unknown origin	5	0
Total	31	24

**Table 3 jcm-11-06227-t003:** Numbers of ambulatory patients in the surgery and non-surgery groups.

	*n* (%)			
	Baseline	1 Month	3 Months	6 Months
Surgery group	15 (48.4)	28 (90.3)	22 (88)	8 (88.9)
Non-surgery Group	17 (70.8)	14 (58.3)	7 (53.8)	1 (50.0)

The values represent the number of ambulatory patients as a percentage of the number of live patients at each timepoint.

**Table 4 jcm-11-06227-t004:** Survival time of patients with a PS of 0–2 versus those with a PS of 3–4.

	*n*	MST (Months)	95% Confidence Interval		*p*-Value
PS 0–2	24	5.17	4.58	5.75	0.004
PS 3–4	31	2.23	1.97	2.50	

MST: median survival time, PS: performance status.

**Table 5 jcm-11-06227-t005:** Survival time of ambulators versus non-ambulators.

	*n*	MST (Months)	95% Confidence Interval		*p*-Value
Ambulators	43	3.53	1.85	5.22	0.009
Non-ambulators	12	2.10	0.52	3.68	

MST: median survival time.

**Table 6 jcm-11-06227-t006:** Survival time of patients with a BI ≥ 60 versus those with a BI < 60.

	*n*	MST (Months)	95% Confidence Interval		*p*-Value
BI ≥ 60	30	4.80	2.74	6.86	0.029
BI < 60	25	2.40	1.21	3.59	

MST: median survival time, BI: Barthel Index.

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
