# Peer review of "Survival Rate after Palliative Surgery Alone for Symptomatic Spinal Metastases: A Prospective Cohort Study"

_jcm, 2022, doi:10.3390/jcm11216227_

Round 1
Reviewer 1 Report
Authors explore the effect of palliative surgery in patients with spinal metastasis.
Just one thing compromising the value of this study is that there is a possibility that the two groups may not have the same clinical course because the primary cancer types and stages of the two groups are heterogenous. This makes a quantitative comparison of the survival rates of the two groups difficult.
If there is data on radiotherapy scheme and local control, it would be good to present it together. Local control associated with ambulatory function may affect survival.
But, this is interesting topic, i expect authors improve the study with additional evaluation.
Author Response
Authors explore the effect of palliative surgery in patients with spinal metastasis.
Just one thing compromising the value of this study is that there is a possibility that the two groups may not have the same clinical course because the primary cancer types and stages of the two groups are heterogenous. This makes a quantitative comparison of the survival rates of the two groups difficult.
If there is data on radiotherapy scheme and local control, it would be good to present it together. Local control associated with ambulatory function may affect survival.
But, this is interesting topic, I expect authors improve the study with additional evaluation.
Answer; first of all, we would like to thank the reviewers for the insightful comments and suggestions. We really appreciate your comment. The radiotherapy was the first choice for treatment of spinal metastasis, because the efficacy and safety were already established by large scales studies. Therefore, all patients received radiotherapy in this study. Non-surgery group was received the radiotherapy within 2 weeks after study commencement. On the other hand, in the surgery group, depending on surgical schedule, 11 patients received preoperatively and 20 patients postoperatively. But these 20 patients received within 2 weeks after study commencement. We described that “The patients received radiotherapy with a mean dose of 29.5 ± 2.9 Gy (range 15 to 34 Gy) within 2 weeks after study commencement.” at line 155-156.

Reviewer 2 Report
The issue proposed by the study is interesting. Thanks to improvements on cancer target drugs, the survival rate of patients is increasing. Symptomatic spinal metastases (SSM) drastically reduce quality of life and the likelihood of continuing cancer chemotherapy. The study’s purpose is to determine whether or not decompressive spinal surgery can improve the clinical outcome in these patients.
The methodological approach is correct and the abstract is complete and exhaustive.
All the data were properly analyzed and described and the conclusion is in line with the obtained results.
The study might be worthy of publication after a minor revision.
- Introduction line 32: the following citation is recommended mentioning possible SRE-related events (DOI: 10.23750/abm.v92iS3.12540)
- A native English speaker might be useful to proofread the text to improve the grammatical content.
Author Response
The issue proposed by the study is interesting. Thanks to improvements on cancer target drugs, the survival rate of patients is increasing. Symptomatic spinal metastases (SSM) drastically reduce quality of life and the likelihood of continuing cancer chemotherapy. The study’s purpose is to determine whether or not decompressive spinal surgery can improve the clinical outcome in these patients.
The methodological approach is correct and the abstract is complete and exhaustive.
All the data were properly analyzed and described and the conclusion is in line with the obtained results.
The study might be worthy of publication after a minor revision.
- Introduction line 32: the following citation is recommended mentioning possible SRE-related events (DOI: 10.23750/abm.v92iS3.12540)
- A native English speaker might be useful to proofread the text to improve the grammatical content.
First of all, we would like to thank the reviewers for the insightful comments and suggestions. We really appreciate your comment. We added the citation what you recommended.
